# Obese Adipocytes Have Altered Redox Homeostasis with Metabolic Consequences

**DOI:** 10.3390/antiox12071449

**Published:** 2023-07-19

**Authors:** Saverio Cinti

**Affiliations:** Scientific Director Centre of Obesity, Marche Polytechnic University, Via Tronto 10a, 60126 Ancona, Italy; saverio.cinti@icloud.com or s.cinti@staff.univpm.it

**Keywords:** adipocyte, obesity, stress, endoplasmic reticulum, ROS, type 2 diabetes

## Abstract

White and brown adipose tissues are organized to form a real organ, the adipose organ, in mice and humans. White adipocytes of obese animals and humans are hypertrophic. This condition is accompanied by a series of organelle alterations and stress of the endoplasmic reticulum. This stress is mainly due to reactive oxygen species activity and accumulation, lending to NLRP3 inflammasome activation. This last causes death of adipocytes by pyroptosis and the formation of large cellular debris that must be removed by macrophages. During their chronic scavenging activity, macrophages produce several secretory products that have collateral consequences, including interference with insulin receptor activity, causing insulin resistance. The latter is accompanied by an increased noradrenergic inhibitory innervation of Langerhans islets with de-differentiation of beta cells and type 2 diabetes. The whitening of brown adipocytes could explain the different critical death size of visceral adipocytes and offer an explanation for the worse clinical consequence of visceral fat accumulation. White to brown transdifferentiation has been proven in mice and humans. Considering the energy-dispersing activity of brown adipose tissue, transdifferentiation opens new therapeutic perspectives for obesity and related disorders.

## 1. Introduction

### 1.1. The Adipose Organ

In mice and humans, white (WAT) and brown (BAT) adipose tissues are organized to form a large adipose organ composed of two compartments, namely subcutaneous and visceral [1,2].

The subcutaneous part is located below the skin and surrounds the whole organism in humans, whilst in mice, it is mainly localized at the root of arms, thus distinguished in the anterior and posterior parts. The visceral part, both in humans and mice, is mainly intra-truncal and surrounds the aorta and its main branches. Furthermore, visceral fat fills all the peritoneal folds, such as the mesentery, mesocolon, mesosigma, greater and smaller epiploon, epiploic appendices, and all other small peritoneal ligaments, including those related to the gynecologic structures [3].

### 1.2. White Adipose Tissue

WAT is the main component of this organ in adults [4]. It is formed by white adipocytes. This cell type is characterized by a single large lipid droplet contained in its cytoplasm (unilocular adipocytes) that is reduced to a thin, barely visible, peripheral rim. The nucleus is squeezed and is shaped as a crescent.

This anatomy is required in order to allow maximal volume for energetic molecules in minimal space (spherical shape) and guarantee their main function: provision of energy to the organism in the intervals between meals [5]. The energy directly provided by adipocytes is in the form of fatty acids that are ready to be used by the most important organ for survival: the heart. But adipocytes, during fasting, are also able to provide the organism with glucose. This happens through an indirect pathway, that is, a glucogenic stimulus to the liver (see below).

The size of adipose organ allows periods of fasting up to several weeks in humans [6].

The adipose organ can be considered as an endocrine organ because it produces important hormones: leptin, asprosin, adiponectin, and several other cytokines [7].

Leptin is secreted in proportion to the size of adipocytes and the amount of WAT [8]. It acts on the brain to push the behavior to search for food when energy storage is scarce. Its functional receptor is present in the neurons of the hypothalamic arcuate nucleus (ARC) that control hunger and satiety, but also in several key areas of the limbic system. High levels of leptin, such as those in obese people, do not reduce food intake, suggesting that leptin resistance develops in parallel with the increased amount of fat. This aspect is in line with the need to accept energy intake, even if storage is high, because a fasting period is, of course, not predictable.

Recent data suggest that ciliary neurotrophic factor (CNTF) could be useful to overcoming leptin resistance [9]. The unexpected weight loss observed in amyotrophic lateral sclerosis patients treated with CNTF attracted the interest of this cytokine for the treatment of obesity [10]. Unfortunately, side effects and the appearance of neutralizing antibodies halted the anti-obesity development of this drug, but further studies revealed that administered CNTF acts through a receptor complex belonging to the IL-6 receptor family that is present not only in the brain, but also in the muscle, liver, and adipose tissue, where CNTF treatment collectively promotes insulin sensitivity and energy expenditure [11].

The anorectic effect of CNTF is thought to be mainly due to a leptin-like activation of JAK2-STAT3 signaling on hypothalamic ARC neurons. However, it has recently been shown that peripheral CNTF administration led to STAT3 phosphorylation mainly in NPY/AgRP neurons that are located in the ventromedial part of the ARC, outside the blood–brain barrier [9]. In addition, it was also shown that it activates ERK signaling in β2-tanycytes, an effect linked to leptin transport into the brain [12]. In line with this hypothesis, by measuring leptin content in hypothalamic protein extracts and quantifying the phosphorylation of STAT3 in mice treated with CNTF and with both CNTF and leptin, it has been shown that CNTF promotes leptin entry and signaling in hypothalamic feeding centers [9]. These data may explain CNTF effectiveness in leptin-resistant obese patients and animals and suggest that investigation of the mechanisms by which CNTF and analogues induce reduction in food intake has the potential to offer novel treatments for morbidly obese leptin-resistant patients [13,14].

Asprosin is secreted by adipocytes during fasting and acts on the liver to stimulate glucogenesis and on the brain to push the behavior of food intake [15,16]. This important hormone was discovered recently when studying the gene responsible for the so-called neonatal progeroid syndrome (NPS) with lypodistrophy. This gene (FBN1) produces an extracellular matrix protein: fibrillin 1. During the process of secretion, profibrillin 1 is proteolytically processed and the C-terminal cleavage product is a 140-amino-acid-secreted polypeptide, named asprosin from the Greek word for white (ασπροσ), because it is abundantly expressed in WAT. Patients with NPS are extremely lean and therefore have very low levels of leptinemia, which should increase food search and intake, but the absence of asprosin strongly reduces food intake.

Adiponectin is secreted by both WAT and BAT and follows the opposite rule to leptin with a negative correlation with fat mass. Its main function is the improvement in insulin sensitivity with anti-atherosclerotic activity on blood vessels [17].

Many other adipokines produced by both white and brown adipocytes have several positive effects, mainly on glucose homeostasis and the cardiovascular and the coagulative systems [7].

### 1.3. Brown Adipose Tissue

Brown adipocytes are polygonal cells smaller that white adipocytes (approximately 40–50 vs. 70–90 microns in diameter, respectively) [18]. Their nucleus is roundish and mainly centrally located. Lipids form small and numerous droplets (multilocular adipocytes). Mitochondria are spherical and packed with laminar cristae [19]. They are provided with a special protein (UCP1), uniquely expressed in this cell type and responsible for their function of thermogenesis. All the intrinsic energy of fatty acids, forming lipid droplets, is dissipated in the form of heat because of the protonophore properties of UCP1. In normal cells, oxidation of substrates produces a proteomic gradient between the external compartment of mitochondria and their matrix. Thus, protons tend to reach the matrix but cannot pass through the membrane and are obliged to transit into the ATPase channel. This proton flux creates energy that is used to convert ADP into ATP. In brown adipocytes, protons can use the protonophore UCP1 instead of ATPase, thus dissipating the intrinsic energy contained in the fatty acids oxidized in the respiratory chain. The multilocular arrangement of lipids in this cell type yields massive lipolysis (occurring only at the surface of lipid droplets), and the large number of molecules oxidized produces physiologically relevant thermogenesis [20].

Thermogenesis is of vital importance because most mammals live in environments below 37 °C, which is the temperature required for normal cellular life and activity.

In mice, BAT is mainly located in the subcutaneous position, specifically in the anterior depot in the interscapular space. A detailed analysis of the whole adipose organ revealed big differences among strains but confirmed the interscapular location as the most important depot, even if several other depots were identified in both the subcutaneous and visceral parts of this organ [21].

In humans, BAT is mainly localized in the visceral part of the adipose organ in a tight relationship with the aorta and its main branches in order to easily transfer heat to blood and therefore to the whole body [2,22]. We recently showed that people living in very cold areas (Siberia) are provided by large amounts of BAT [23].

Recently, it has been observed that brown adipose tissue also produces different types of factors that have hormone-like functions. Overall, these factors are called batokine and are mainly of a peptide and lipid nature. The main ones are growth factor FGF21, neuregulin 4, IL-6, adiponectin, myostatin, lipokine 12,13diHOME, and miR-99b [24]. Overall, all these factors have positive metabolic effects and an increase in cardiovascular efficiency. The effect of mutual functional reinforcement between brown adipose tissue and skeletal muscle is particularly interesting. In fact, the latter, when activated, produces irisin [25], which has a positive stimulating effect on brown adipose tissue [26].

A new cell of the brown adipose tissue parenchyma has recently been discovered. This new cell type has a morphology very similar to that of brown adipocytes but with special mitochondria that do not express UCP1. Its functional role would be to produce acetate capable of regulating the production of heat by brown adipose tissue [27]. In fact, the “Harlequin effect” that occurs in this tissue when it is acutely activated has been known for decades; with the immunohistochemistry for UCP1, intensely reactive cells are highlighted next to cells completely switched off from a functional point of view, which produce heterogeneous staining that recalls the famous mask [28]. Together with the above-mentioned acetate, the data argue that acute UCP1 activation also results in the production of other functional inhibitors that likely protect cells from undesirable heat shock effects [29].

### 1.4. Transdifferentiation Properties of the Adipose Organ

In most organs, tissues with different physiology cooperate for a common finalistic purpose. For example, the stomach muscles produce peristalsis, while mucosae produce gastric juice. Their cooperation is evident for digestion. As such, we asked what cooperation exists between WAT and BAT. After several experiments, our data support the idea that physiologic stimuli could induce reversible transdifferentiation between WAT and BAT [30]. During chronic cold exposure, WAT can transdifferentiate (direct conversion) into BAT to increase thermogenesis, and during chronic energy positive balance, BAT converts to WAT in order to increase the energy-storage capacity of the organ. Recently, this capacity of reciprocal and reversible conversion was also proved by the powerful lineage-tracing technique [31].

Thus, transdifferentiation, i.e., the capacity of cells to reprogram their genome in order to directly convert a mature cell into another cell type with different anatomy and physiology, seems to be the important cooperation of WAT and BAT in the adipose organ.

### 1.5. Another Example of Transdifferentiation in the Adipose Organ

This innovative theory about the basic properties of cells deserves other examples and an in-deep study.

In female mice, subcutaneous fat forms mammary glands during pregnancy and lactation. The cells producing milk (alveolar cells) are not normal constituents of breast outside pregnancy and lactation. They develop during pregnancy and lactation. In virgin mice, the breast is composed of a mixture of WAT and BAT infiltrated by branched mammary ducts ending in a nipple. Mice have five bilateral nipples, thus a total of ten potential mammary glands develop during pregnancy and lactation. In these periods, epithelial alveolar glands develop in parallel with the disappearance of adipocytes [32].

Several experiments, including electron microscopy, BrdU data, and explants, suggested adipo–epithelial transdifferentiation during pregnancy and lactation and epithelial–adipo transdifferentiation in the post-lactation period (reviewed in [33]).

Our lineage-tracing data [33] strongly supported the hypothesis, even if some authors deny the phenomenon based only on their lineage-tracing data [34]. We recently confirmed epithelial–brown transdifferentiation in the post-lactation period by lineage tracing [35]. Interestingly, data from another independent group also showed that transdifferentiation phenomena (adipo–myoepithelial and myoepithelial–adipo) can occur in mammary glands in pregnancy and post-lactation periods [36].

## 2. The Obese Adipose Organ

Obesity is a well-defined clinical condition. BMI is an index mainly related to body fat, expressing the relationship between height and weight. It is used to classify individuals as underweight (<18.5), normal (18.5–24.9), overweight (25–29.9), or obese (>30) [37]. A total of 90% of type 2 diabetes patients are obese or overweight [38]. Thus, a strict relationship between fat accumulation and type 2 diabetes seems to exist.

The early work produced mainly by Bruce Spiegelman and Gokan Hotamisligil outlined the importance of TNFα in the adipose tissue of mice and humans [39]. They showed that this factor was increased in obese fat and responsible for insulin resistance, thus providing the first link between obesity and glucose dysmetabolism. Reinforcing the negative metabolic effects, later data showed a role of TNFα in the necrosis of brown adipocytes [40,41].

In this review, an attempt has been made to outline how obesity modifies the anatomy and molecular pathways of adipocytes in the adipose organ and how these alterations can explain, at least in part, the well-evident clinical link between obesity and type 2 diabetes.

In 2003, two independent laboratories published a milestone observation at the same time: obese fat of mice and humans is characterized by low-grade inflammation, mainly sustained by macrophages [42,43]. Furthermore, they showed that TNFα, IL-6, and I-NOS are mainly produced by macrophages and their increase in the blood precedes the onset of insulin resistance.

We showed that the cause of inflammation is the death of adipocytes. Starting from the observation that more than 90% of macrophages are restricted to characteristic histopathology figures, we pointed to the study of these aspects to try to understand the underlining phenomena ending in the inflammation. Macrophages formed lipid-associated structures we called crown-like structures (CLSs) [44].

CLSs were found also in lean fat, but obese fat was enriched by about 30 times, thus supporting inflammation. Each CLS was formed by a central large lipid droplet, mimicking a normal adipocyte, surrounded by several (about 20–30 in a section) macrophages immunoreactive for MAC2, thus indicating an active phagocytic stage [45]. Perilipin 1 plays a key role in lipolysis; thus, it is essential for adipocyte survival. Immunostaining for perilipin 1 (PLIN1) revealed that the lipid droplets surrounded by macrophages were PLIN1-negative, thus suggesting the death of adipocytes. Electron microscopy confirmed the death of adipocytes, showing remnants such as irregular and altered basal membrane invaded by macrophages and cytoplasmic debris in phagosomes into the cytoplasm of macrophages. Furthermore, active lipid phagocytosis was evident at the side of macrophages in contact with the lipid droplet. All together, these observations support a time course of events: obese adipocytes die, and macrophages come to their natural role of scavenging debris derived from dead cells. In support of this idea is also the frequent presence of giant multinucleated macrophages, typical of the histopathology of foreign body reaction, in CLSs.

The discovery of a lifetime for normal adipocytes fully explained the presence of CLSs in lean adipose tissue [46]. Notably, knockout of the hormone-sensitive lipase (HSL), a major lipase in mature adipocytes, resulted in increased lipid storage and adipocyte hypertrophy but not increased fat mass or obesity [47]. In these lean mice, we found typical CLSs with a density like those found in obese animals, strongly suggesting that death is due to adipocyte hypertrophy and not to obesity [44].

More recent results also fully confirmed the proposed time course. We used a transgenic mouse model in which the death of adipocytes specifically can be induced by administration of a dimerizer that causes caspase-8 activation and the death of adipocytes. We showed that after the administration of the dimerizer to mice, there was a progressive increase in the number of dead adipocytes in their adipose tissues and subsequent parallel development in the number of CLSs. Furthermore, we observed that all dead adipocytes formed CLSs [48].

### 2.1. Molecular Links between Fat Chronic Inflammation and Insulin Resistance

Since the early work of Spiegelman and Hotamisligil, a molecular link between the adverse effects of TNFα on insulin receptors (IRs) has been identified [39]. Inhibition of tyrosine phosphorylation in substrate 1 of the insulin receptor (IRS-1) caused reduced IR activity, resulting in insulin resistance. Thus, fat chronic inflammation seemed to be linked to insulin resistance, which is a well-known clinical aspect strongly correlated to BMI [49]. Insulin resistance due to fat inflammation extends to skeletal muscles and liver, thus implying important pancreatic activity for hormone production [50]. Experimental data from mice and humans recently showed a sort of adaptive response of Langerhans islets to this hyper-insulinemic situation. As a matter of fact, a progressive increase in density of inhibitory noradrenergic fibers was found in ob/ob and db/db obese Langerhans islets of mice in parallel with their progression in obesity [51]. It is worth noting that electron microscopy showed an increased number of synaptoid contacts with beta cells. This kind of adaptive response seems to be responsible for beta cell stress and hypo-functionality that can be related to the clinical onset of type 2 diabetes [52,53].

Thus, chronic fat inflammation could link obesity to type 2 diabetes.

### 2.2. The Cause of Death of Hypertrophic Adipocytes

The main alteration visible in obese fat is the hypertrophy of white adipocytes. In genetically obese mice, the size of obese adipocytes is about 7–11 times that of lean adipocytes [54]. In humans, the increase is about 2–3 [55]. This new anatomy implies an increased distance from capillaries and consequent hypoxia, as demonstrated by HIFα production [56]. Furthermore, it is well known that an excess of nutrients can cause, per se, endoplasmic reticulum stress in adipocytes [57]. Thus, it is not surprising that electron microscopy can reveal signs of stress and alterations of organelles in hypertrophic adipocytes [58].

The normal anatomy of adipocytes suggests that their expansion, in the case of an excess of nutrients, must be limited. As a matter of fact, different types of collagens surround the external surface of lean adipocytes. A thin amorphous envelope of collagen IV (basal or external lamina) surrounds the cell from the early stages of development. Transmission (TEM) and high-resolution scanning electron microscopy (HRSEM) recently showed that a thin network of collagen fibrils (with the size of type III collagen) forms a further external reinforcement that is likely devoted to limiting any excess of expansion [58]. In obese adipocytes both structures (basal membrane and fibrils network) look reinforced with increased thickness of the first [59] and increased size (unpublished) and number of fibrils [58] suggesting a sort of reaction of the cell to hypertrophy. Together with these reactions, c-jun N-terminal kinase (JNK) and nuclear factor kB (NFkB) signaling are upregulated, leading to increased expression of downstream cytokines, such as TNF-α, IL-6, and monocyte chemoattractant protein-1 MCP-1 and HP [60,61]. These last two are potent chemoattractants for macrophages, thus leading to preliminary inflammatory invasion surrounding moribund stressed hypertrophic adipocytes and ending in CLS formation after the death of the adipocytes (Figure 1).

Electron microscopy revealed that the endoplasmic reticulum (ER) (Figure 2) and mitochondria appear as the most altered organelles in stressed adipocytes [58]. The ER was dilated, and the mitochondria reduced in number and size [58].

### 2.3. Molecular Mechanisms Responsible for the Oxidative Stress Induced by High Levels of Lipids in Obese Adipocytes

The endoplasmic reticulum (ER) is a membrane-bound organelle that is responsible for the folding, modification, and synthesis of secretory and structural proteins. The process of protein synthesis and folding is highly controlled and is sensitive to the perturbation of ER homeostasis. The unfolded protein response (UPR) refers to the mechanisms by which cells control ER protein homeostasis.

Mitogen-activated protein kinases (MAPKs) are involved in several cellular responses to a series of diverse intra- and extracellular stimuli.

Hypertrophy of adipocytes caused by obesity induce both the UPR and stress-activated MAPK pathways (reviewed in [57]).

UPR signaling is initiated by three ER-membrane-associated proteins: PERK (PKR-like eukaryotic initiation factor 2a [eIF2a] kinase); IRE1 (inositol-requiring enzyme 1); and ATF6 (activating transcription factor 6). They act in a concerted way to control protein synthesis and the degradation and production of molecules necessary for ER homeostasis. Several chaperons were identified as responsible for the adaptive ER responses, and the ER-resident leucine-zipper transcription factors ATF6 and its downstream-activated protein XBP1 stimulate the expression of a broad series of genes involved in several pathways, such as protein folding, secretion, and degradation, to clear misfolded proteins from the ER. PERK is responsible for translational reduction in the arrival of new proteins in the ER.

The UPR undertakes many critical processes for normal cellular homeostasis, such as survival pathways, immune responses, and nutrient sensing. In each case, the UPR activates pathways to control processes, which can have effects both within and beyond the cell in which they are activated. Prolonged activation of distinct signaling networks stimulated by each branch of the UPR is a critical cause for adverse outcomes [60].

Under ER stress conditions, such as those presented by nutrient overload, the responses induced by ER stress result in a maladaptive set of events, leading to various cellular or systemic pathologies, including death by pyroptosis [58]. In particular, PERK and ATF6 induce expression of transcription factor C/EBP homologous protein (CHOP), which in turn leads to reduced expression of the antiapoptotic gene Bcl2 and increased expression of a number of proapoptotic genes. Nutrient-sensing pathways are linked to ER function, for example, the mTORC1 complex, which is a major cellular nutrient sensor that is coupled to the UPR that is thus sensitive to the nutritional status of the cell, responding to glucose deprivation, exposure to excess fatty acids, hypoxia, and growth stimuli. Importantly, all three components of UPR regulate immune signaling NF-kB during ER stress. Furthermore, signaling through toll-like receptors (TLRs) can activate IRE via ROS with inflammatory cytokine production. Activation of XBP1 by IRE1 promotes the sustained production of inflammatory mediators, including interleukin (IL) 6.

Furthermore, experimentally induced ER stress in cultured cells can cause increased expressions of many inflammatory molecules, such as IL8, IL6, MCP1, and tumor necrosis factor (TNF) [57].

Importantly, cytokines and their pathways can influence the function of ER. This is negatively influenced by pathways involving JNK and IkB kinase (IKK) and by some mediators, such as ROS. The duration and level of oxidative stress and the levels and magnitude of ROS and/or NO production can induce a maladaptive balance in ER responses.

JNK and p38 MAP kinases respond to inflammatory cytokines and to many chemical and physical changes in the environment, as well as DNA damage and redox imbalance, thus they are often called stress-activated MAPKs.

Stress-activated MAPKs play a key role in the regulation of inflammatory cytokine expression. They target transcription factors and chromatin-remodeling enzymes that regulate the expression of cytokines during inflammatory responses.

The immune response due to inflammation is strictly related to the involvement of specific innate and adaptive immune cells. These types of cells are influenced by stress- activated MAPK signaling pathways. Macrophages represent an important part of the immune response often described as innate. Stress-activated MAPKs can control macrophage polarization, specifically as M1 (associated with the expression of inflammatory cytokines) or M2 (expressing anti-inflammatory cytokines), and are involved in tissue remodeling. JNK is required for the M1, but not M2, macrophage phenotype.

Feeding a high-fat diet causes metabolic stress and activates stress-activated MAPK signaling pathways. This is mediated by increased amounts of saturated free fatty acids that activate the MLK group of MAPKKKs, which are responsible for the canonical phosphorylation-dependent activation of MAPK [57].

As a consequence of most of the above-cited pathways, in obese adipocytes, reactive oxygen species (ROS) are produced and accumulated in the ER (hydrogen peroxide and other peroxides, superoxide radical, hydroperoxyl radical, and hydroxyl radical), reaching levels that can be toxic to the cell [62,63].

Another kinase, PKR (protein kinase RNA-activated), is activated during ER stress, leading to the activation of NLRP3 inflammasome [64,65]. In addition, we showed the presence of cholesterol crystals in obese adipocytes [58], in line with the well-known cholesterol level increase in adipocytes in parallel with their size [66]. This further reinforces the idea that stressed adipocytes are able to activate inflammasomes because cholesterol crystals have been shown to trigger NLRP3 inflammasomes in synovial tissue, causing gout inflammation at this site [67]. Furthermore, many cholesterol crystals were also found in the macrophages of CLSs, reinforcing the idea regarding their clearing activity. Thus, hypertrophic-stressed adipocytes have several inflammasome inductors.

It is worth noting that stressed adipocytes, through the autocrine stimulus of IL-1β, also produce a protein, namely sFRP4 (secreted frizzled-related protein 4), acting on pancreatic β-cells with the inhibition of insulin secretion, further contributing to the dysregulation of the glucose metabolism [68].

NLRP3 is a complex structure composed of a sensor (NOD-like receptor), an adaptor (ACS), and an effector (Caspase-1) [69]. Its activation produces proinflammatory IL-1β and IL-18 [70]. Our data showed the increased gene expression of NLRP3, ASC, and Caspase 1 in the obese fat of mice (both genetically obese and diet-induced by HFD); furthermore, we also showed an increased expression of thioredoxin (TRX)-interacting protein (TXNIP), which induces NLRP3 inflammasome activation following oxidative stress. Immunohistochemistry showed the presence of NLRP3, ASC, and Caspase 1 in the cytoplasm of hypertrophic adipocytes and in CLS macrophages, but not in those distant from CLS, suggesting that the proteins into macrophages are derived from the debris of adipocytes. It is worth noting that we did not find the inflammasome proteins in either the fat of control lean mice or in the adipocytes or CLS of FAT-ATTAC mice, a transgenic model of apoptotic adipocyte cell death [58].

Thus, stressed hypertrophic adipocytes produce and secrete IL-1β and IL-18 due to inflammasome activation. IL-1β is elevated in the blood of obese mice and humans with strong implications for insulin-secretion inhibition as reported above [70,71], as well as for the disruption of insulin signaling in peripheral tissues [72]. Notably, blockade of IL-1β by the specific human receptor antagonist anakinra in patients with type 2 diabetes leads to significantly improved glycemic control and reduced inflammation. Furthermore, mice lacking NLRP3, ASC, or Caspase-1 are resistant to the development of high-fat-diet-induced obesity and insulin resistance [73]. Caspase-1 is a cysteine protease able to induce a form of cell death known as pyroptosis [70]. This type of cell death is characterized by rupture of the plasmalemma, water influx, cellular swelling, osmotic lysis, and release of proinflammatory cellular content. This type of cell death is different from apoptosis and is related to inflammasome stimulation [74].

### 2.4. The Concept of Critical Death Size

A very well-known clinical aspect of obesity is the striking difference between subcutaneous and visceral obesity. Since the seminal work of Jean Vague, it has been clear that the metabolic consequences of obesity and especially type 2 diabetes are strictly linked mainly to visceral fat accumulation [75]. This concept was later fully confirmed by other authors [76].

Furthermore, it was recently shown that visceral fat accumulation correlates with higher mortality and incidence of cardiovascular diseases, even in subjects with a normal BMI [77].

The anatomical well-known difference between subcutaneous and visceral fat consists mainly of their size, i.e., smaller in visceral fat [78]. Furthermore, a higher vascular and parenchymal-nerve-fiber density in visceral fat was also found [79,80]. The reason for these differences is not known; however, considering that a large proportion of visceral fat is BAT in infants [81] and in people affected by pheochromocytoma or living in cold countries [22,23], it could be speculated that the visceral fat of adults is derived from a BAT–WAT conversion due to the normal reduced activity of the sympathetic nervous system with age [82]. Thus, considering that brown adipocytes are smaller than white adipocytes, visceral fat could be composed of smaller cells because it is BAT-derived.

As a matter of fact, visceral fat is more prone to death when requested to enlarge, such as in obesity. By comparing the size of visceral and subcutaneous adipocytes in obese mice and humans, we [54,55] and other authors [78,83] found larger subcutaneous than visceral adipocytes.

Considering the positive correlation existing between the size of adipocytes and the number of inflammatory macrophages in the tissue [43], we expected to find a higher number of CLSs in subcutaneous fat, but the reality was the reverse [54,55]. This observation raised the concept of critical death size. When exposed to a positive energy balance, such as in obesity, the smaller visceral adipocytes cannot enlarge as subcutaneous adipocytes and die at a smaller size than subcutaneous adipocytes [84].

Further data supporting this hypothesis that BAT-derived small visceral adipocytes have a lower critical death size than subcutaneous adipocytes come from our recent experiments [85]. Studying mice lacking adipose triglyceride lipase (ATGL), we observed that all brown adipocytes convert from multilocular into unilocular cells. Subcutaneous adipocytes reacted with hypertrophy even if mice were lean. The size reached by converted brown adipocytes and hypertrophic white adipocytes was similar, but the number of CLSs was significantly higher in BAT-converted visceral fat, strongly supporting the idea that visceral adipocytes have a smaller critical death size because they are derived from BAT conversion. Interestingly, by comparing the number of CLSs in BAT-converted fat in different anatomical locations, we observed a higher number of CLSs in interscapular fat than in mediastinal fat. Interscapular fat is the classic BAT with a higher density of parenchymal nerve fibers, and thus is the physiologically most important BAT for thermogenesis. This suggests that the stricter brown phenotype plays a role in the determination of critical death size [85].

## 3. Conclusions

The new concept of adipose organs and the new data supporting its validity for humans has a lot of clinical implications. Starting from the relationship between obesity and type 2 diabetes [38], it’s getting clearer that the redox state of the parenchymal cells of this organ play a key role in the development of mechanisms, resulting in several aspects of the dysmetabolism typical of metabolic syndrome. Here, we focused only on the mechanisms dealing with insulin resistance, but several other consequences of obese adipose organs connect with other alterations of this syndrome, such as hypertension and atherosclerosis [86,87].

The presence of a second type of adipose tissue (BAT) in the adipose organ is of paramount importance not only for the theoretical aspects of organ definition, but also for several angles of pathophysiology and for therapeutic perspectives, starting from the discovery of transdifferentiation.

The pathophysiology aspects include BAT–WAT conversion with critical death size implications and the reversible adipo–epithelial conversion in the mammary glands during pregnancy and lactation in females, with possible implications for a better understanding of oncologic phenomena [32].

The therapeutic perspective includes WAT–BAT conversion that has already-proven efficacies in treating obesity in small mammals [88,89,90,91,92] and that could be studied in humans where BAT activity has been proven efficient for health [93], also considering that BAT-agonist drugs for humans are now available [94,95,96].

The take-home message of this review is that humans are provided with a real new organ, overtaking the old idea of adipose tissues (WAT and BAT) as separated entities without any functional interconnections. This new concept implies that mature cells can reprogram their genome and change their phenotype and physiology (transdifferentiation) with a series of practical consequences from pathophysiology interpretations, including a possible explanation as to why visceral fat accumulation is more unhealthy than subcutaneous fat accumulation, for new strategies for future therapies, such as browning the adipose organ.

## Figures and Tables

**Figure 1 antioxidants-12-01449-f001:**
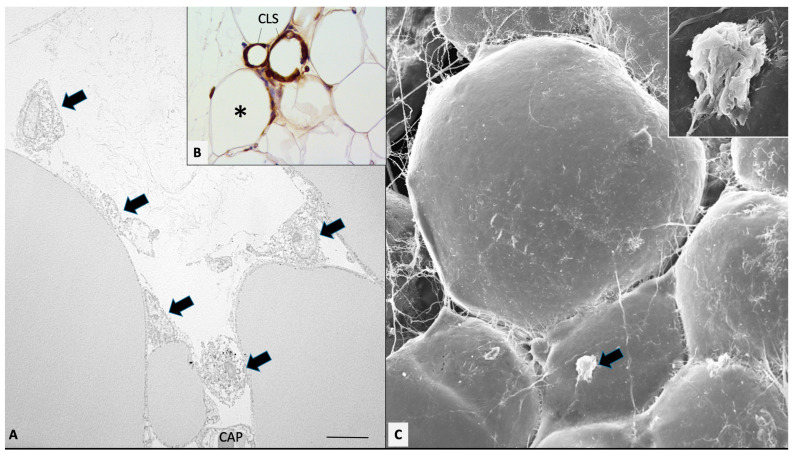
Mesenteric fat of obese db/db mice. Transmission (**A**) and high-resolution scanning electron microscopy (**C**) showing early stage of macrophages invasion (arrows). (**B**): MAC2 immunohistochemistry showing both early (asterisk) and late stages of macrophage invasion, ending in CLS formation (as indicated). In (**C**), the indicated macrophage is enlarged in the inset. CAP: capillary. Bar: 8 μm in (**A**), 80 μm in (**B**), and 20 μm in (**C**) (2 μm in inset).

**Figure 2 antioxidants-12-01449-f002:**
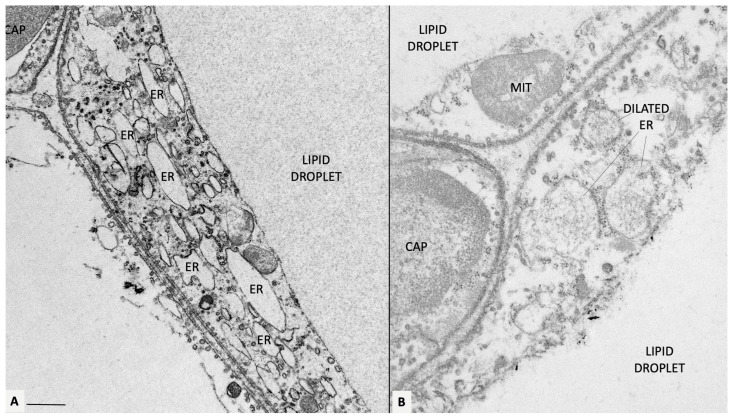
Mesenteric fat of obese db/db mice. (**A**) Representative transmission electron microscopy of obese adipocytes. Endoplasmic reticulum is diffusely dilated (ER, some indicated). In (**B**), a fine microstructure network is visible in the ER lumen, possibly corresponding to accumulated ROS. Original figures by the author. CAP: capillary; MIT: mitochondrion. Bar: 0.4 μm in (**A**) and 0.6 μm in (**B**).

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
