# Peer review of "Obese Adipocytes Have Altered Redox Homeostasis with Metabolic Consequences"

_antioxidants, 2023, doi:10.3390/antiox12071449_

Round 1
Reviewer 1 Report
The paper of Cinti analyzes the dysfunctions of adipose tissue cells in relation to oxidative stress induced in obesity. Although there are some novelties, such as the evaluation of CLS in the adipose tissue of the obese subjects and their role in the death of adipose cells, overall, the topics covered have already been reported by other authors. Furthermore, the title underlines the relationship between redox homeostasis and metabolic consequences. However, these aspects are not analyzed in detail in the review.
Thus, in my opinion it is therefore advisable to describe in details:
1. the mechanisms responsible for the oxidative stress induced by high levels of lipids in adipocytes
2. how oxidative stress is correlated with an alteration of the metabolism of adipose cells or other districts (hepatic/muscular).
Reviewer 2 Report
In this review article, the author summarizes the alterations and pathology of adipocytes observed in the obese state at biological, molecular and clinical level. Although convincing reference articles and rich information successfully support each subject discussed in the manuscript, it is not clear what is the central message of the current article. The points outlined below need to be addressed.
Major comments:
(1) Structure of the manuscript needs to be improved.
· The author needs to add an ‘Introduction’ session, where background including significance, recent advances, and critical issues, and the main focus the review article discusses need to be clearly mentioned. Due to a lack of these descriptions, it is difficult to follow the current form of the manuscript.
· It would be more appropriate to have main- and sub-paragraphs according to the context throughout the manuscript.
· ‘Take-home messages’ in the manuscript need to be distinctly described in a 'Conclusion' session. Further, it is highly recommended to include future directions in the field of adipocyte research in obese states in the same session.
(2) Although the current title describes 'Obese adipocytes have altered redox homeostasis with metabolic consequence', it is not clearcutting how dysregulated redox homeostasis underlies all subjects discussed in the article, particularly white to brown trans-differentiation of adipocytes. It could be assumed that altered redox homeostasis is the initial molecular event that occurs in obese adipocytes, leading to local and systemic pathology. This point should be carefully described in order to lead to the author's conclusion.
(3) By now, it is recognized that brown adipose tissue is also a secretory organ of autocrine and paracrine factors, contributing to systemic metabolism. This point should be included in the manuscript.
Minor comments:
(1) Figure legends for Figures 1 and 2 are not precise. Bars in Figure 1C and Figure 2 B are not indicated in the images. Figure 2A is missing the label. Further, if the images are original data obtained by the author, that point needs to be mentioned in the legend. Otherwise, the references to the data should be mentioned.
Moderate editing of English language required
Round 2
Reviewer 1 Report
The paper has been modified in accordance with the reviewer's requests. THus is now accetable for publication on Antioxidants